# HSDirSniper: A New Attack Exploiting Vulnerabilities in Tor's Hidden Service Directories

## ABSTRACT

Tor hidden services (HSs) are used to provide anonymous services to users on the Internet without revealing the location of the servers, enabling freedom of speech. However, existing approaches have proven ineffective in mitigating the misuse of hidden services. Our investigation reveals that the latest iteration of Tor hidden services still exhibits vulnerabilities related to Hidden Service Directories (HSDirs). Building upon this identified weakness, we introduce the HSDirSniper attack, which leverages a substantial volume of descriptors to inundate the HSDir's descriptor cache. This results in the HSDir purging all stored descriptors, thereby blocking arbitrary hidden services. Notably, our attack represents the most practical means of blocking hidden services within the current high-adversarial context. The advantage of the HSDirSniper attack lies in its covert nature, as the targeted hidden service remains unaware of the attack. Additionally, the successful execution of this attack does not require the introduction of a colluding routing node within the Tor Network. We conducted comprehensive experiments in both real-world Tor Network environments and simulated settings, and the experimental results show that an attacker equipped with a certain quantity of hidden servers can render arbitrary hidden services inaccessible up to 90% of the time. To ascertain the potential scope of damage that the HSDirSniper attack can inflict upon hidden services, we provide a formal analytical framework for quantifying the cost of the HSDirSniper attack. Finally, we discuss countermeasures and future work.

## CCS CONCEPTS

• **Computer systems organization** → **Embedded systems**; *Redundancy*; Robotics; • **Networks** → Network reliability.

## KEYWORDS

Anonymous communications, HSDir, DoS attack, Tor hidden services

ACM Reference Format:

Anonymous Author(s). 2018. HSDirSniper: A New Attack Exploiting Vulnerabilities in Tor's Hidden Service Directories. In *Proceedings of Make sure to enter the correct conference title from your rights confirmation emai (WWW '24)*. ACM, New York, NY, USA, 12 pages. https://doi.org/XXXXXXX.XXXXXXX

## 1 INTRODUCTION

Tor [6] is one of the most popular anonymous communication networks designed to safeguard the anonymity of both senders and recipients. The hidden service (HS) is a mechanism developed by Tor to ensure receiver anonymity by concealing the IP address of the service through an exposed onion address. However, as the anonymity provided by the HS was available to everyone, it quickly became an accomplice to cybercrime [25], such as botnet servers [4], phishing websites [26], and illegal black markets [24].

To mitigate the issue of hidden service abuse, several methods have been suggested by researchers, only focusing on hidden service denial-of-service (DoS) attacks. Given the inherent characteristics of the hidden services protocol, there exist three key components in the communication path between the client and the hidden services, namely *the hidden service itself*, *the guard relay*, and *the hidden service directory server* (HSDir). As a consequence, the aforementioned three components emerge as the principal focal points for DoS attacks on hidden services.

First, **DoS attacks against the hidden service itself** exploit the vulnerability arising from the asymmetry inherent in the hidden service protocol. For instance, Rochet et al. [15] demonstrated in their work that attackers can degrade the quality of service provided by a hidden service by consuming its machine resources. However, the recent advancements in Tor's system architecture optimization and the introduction of *Onionbalance* [20] have rendered the implementation of DoS attacks against hidden services ineffective as a solution to the problem of hidden service abuse [14].

Second, **DoS attacks against the guard relay of the hidden service** exploit the fact that the guard relay serves as the sole entry point for the hidden service within the Tor Network. However, successful execution of this attack necessitates resolving the practical challenges which the attacker must ascertain the guard relay of the hidden service. While Many studies [5, 13, 15] have demonstrated the feasibility of guard relay discovery attacks, the Tor Project has developed the VanGuard mechanism [17] to counteract them, causing the attack to fail.

Third, **attacking the responsible HSDirs of the hidden services** is an easily overlooked approach. The descriptor encompasses the contact information for the hidden service, which is stored within a designated group of nodes referred to as responsible HSDirs, selected by the hidden service from the Distributed Hash Table (DHT). Tan et al. [16] have taken advantage of the predictability of responsible HSDirs to impersonate them. Nonetheless, this approach exhibits a non-negligible failure rate when applied within real Tor Network. Its effectiveness is contingent upon the relative stability of the DHT [18], and the addition of a new HSDir may thwart the successful implantation of the attacker's node into the intended location, thereby resulting in a failure of the attack. Moreover, the latest iteration of Hidden Services addresses the vulnerability associated with the predictability of responsible HSDir, thus rendering Tan's approach ineffective.

In this paper, we are the first to discover two protocol vulnerabilities in HSDir for storing and cleaning descriptors: (1) the HSDir fails to detect whether it should receive a descriptor for a hidden service or not. This flaw allows the attacker to create hidden services at a relatively low cost, deceiving the victim HSDir into receiving the descriptors of the attacker's hidden services; (2) when the HSDir cleans up the descriptor cache, it employs a descriptor-aging mechanism that removes old descriptors at an hourly time granularity. Consequently, if an attacker manages to flood the HSDir's descriptor cache with malicious descriptors within an hour, the HSDir will purge all stored descriptors.

Leveraging the aforementioned vulnerabilities, we introduce the **HSDirSniper** attack, which blocks arbitrary clients from contacting the target hidden service by forcing the responsible HSDirs to clear the descriptor of the target hidden service. In contrast to the methodology of directly targeting the hidden service itself [15], our approach remains imperceptible to the target hidden service. The primary consequence of our attack is the imposition of an access timeout on any client, rather than augmenting their access duration. When juxtaposed with the work of Tan et al. [16], our attack exhibits superior adaptability to the dynamic and variable landscape of DHT within real Tor Network.

In addition, evaluating the cost of an HSDirSniper attack is crucial for helping attackers allocate their attack resources wisely. In our attack, the number of responsible HSDirs and their descriptor caching thresholds are two important parameters for calculating the cost. However, due to the complexity of hidden services in selecting responsible HSDirs, there is no tool or scheme that can directly determine the number of responsible HSDirs. To fill the gap, we are the first to propose a general theoretical framework for estimating the number of responsible HSDirs.

Empirically, we have designed comprehensive experiments to lunch the HSDirSniper attack and validated the correctness of this theoretical cost estimation framework. We also discuss the effectiveness of HSDirSniper and the countermeasures of HSDirSniper . Our major contributions include: (1) A practical DoS attack capable of blocking arbitrary Tor HSs; (2) The first formal analysis framework for evaluating the cost of launching HSDirSniper attack on multiple hidden services; (3) Suggestions on how to defend the DoS attacks against the HSDir to make Tor HS more robust.

## 2 BACKGROUND

In this section, we delve into the publication process of hidden service descriptors, the principles governing HSDir storage of descriptors. The hidden service mechanism [22] is described in Section 8.1.

### 2.1 Publishing hidden service descriptors

To guarantee continuous availability, the V3 hidden service adopts a two-pronged descriptor approach, encompassing both "current descriptor" and "next descriptor". Each descriptor, uniquely identified by its descriptor ID, is replicated, thereby generating two replicas. This distinction is exemplified in Eq. 1:

$$desc\_id = H(\text{"}store - at - idx\text{''}|K_b|replica|TL|TP) \quad (1)$$

where H symbolizes the SHA256 hash function, with "|" representing the splice symbol. TL is determined by the consensus parameter "hsdir-interval", defaulting to 1440, equivalent to one day. TP signifies the count of TL intervals since 1970-01-01 12:00:00 (valid-after). The blinded public key of the hidden service, $K_b$, is derived from the master identity public key $K_p$ and the time period TP. The descriptor's replica number is indicated by replica, usually set to either 1 or 2. Consequently, a hidden service boasts four distinct replicas: $desc\_id1_{current}$, $desc\_id2_{current}$, $desc\_id1_{next}$ and $desc\_id2_{next}$. To facilitate descriptor publication, the hidden service leverages its consensus file, constructing a distributed hash table (DHT) to pinpoint the appropriate HSDirs. Each HSDir's index on the DHT is determined as:

$$hsdir\_index(node) = H(\text{"}node - idx\text{"}|K_p|SRV|TP|TL) \quad (2)$$

where SRV denotes the shared random value, refreshed and published in the consensus file daily at 00:00, with a 24-hour lifespan.

As shown in Fig. 8a (See Appendices 8.4), while every hidden service concurrently manages two descriptors, it also computes two separate DHTs ($DHT_{current}$ and $DHT_{next}$). The "current descriptor" is allocated to $DHT_{current}$, whereas the "next descriptor" finds its place in $DHT_{next}$.

To ensure high availability of descriptors, the hidden service dispatches each replica to four successive responsible HSDirs. As a result, the total number of responsible HSDirs for each hidden service stands at $4 \times 4 = 16$.

To maintain descriptor validity, the hidden service integrates a descriptor re-upload mechanism. An analysis of the hidden services protocol reveals two primary triggers prompting descriptor re-uploads: (1) **Time-based**: A periodic re-upload typically occurring 60-120 minutes post the preceding upload; (2) **Circuit issues**: A disruption in the HS-IP circuit necessitates a descriptor rebuild.

## 2.2 Cleaning Hidden Service Descriptors

Upon receiving and checking a descriptor, the HSDir generates a structure for its storage. This structure encompasses the super-encrypted segment of the descriptor (which houses IPOs), the descriptor's lifespan, the blinded public key, and the complete descriptor. Notably, since the super-encrypted portion is stored twice, the memory imprint of the stored descriptor marginally exceeds its actual size. The blinded public key is leveraged as a unique identifier for the stored structure.

Descriptors in HSDir storage are subjected to cleanup under three circumstances: (1) **Expiry**: Descriptors have a finite lifespan, as stipulated within the descriptor itself (defaulting to 3 hours); (2) **Overwrite**: A newer descriptor from the same hidden service replaces the existing one; (3) **Memory Constraints**:When the descriptor cache overshoots the defined threshold (MaxMemInQueues), HSDir employs a descriptor aging mechanism. This purges the oldest descriptors in 1-hour increments until the utilized memory drops below $0.2 \times$ MaxMemInQueues.

To fortify against memory-centric attacks, such as the sniper attack [11], Tor incorporates a memory management module. Within this module is the MaxMemInQueues configuration, empowering node operators to dictate Tor's maximum memory utilization. By

default, the value of MaxMemInQueues (expressed as Q) is given by

$$
Q = \begin{cases} 0.75 \times m, & m \leq 8G \\ 0.4 \times m, & 8GB < m \leq 20GB \\ 8GB, & 20GB < m \end{cases} \tag{3}
$$

where m indicates the memory of the host.

## 3 METHOD

In this section, we delve into the intricacies of the HSDirSniper attack, elucidating its operational mechanics, and further develop a theoretical framework to evaluate the cost of the attack.

### 3.1 The Mechanism of the HSDirSniper Attack

The HSDirSniper is a potent assault, designed to incapacitate any V3 hidden service by zeroing in on the responsible HSDirs linked to the target hidden service. The efficacy stems from two fundamental vulnerabilities inherent to HSDir: (1) **Indiscriminate Descriptor Acceptance**: The HSDir does not detect whether it should receive a descriptor for a hidden service or not, leading it to store a substantial number of malicious descriptors; (2) **Unsophisticated Aging Mechanism**: The aging mechanism employed by the HSDir cleanup descriptor is overly simplistic, rendering it vulnerable to exploitation by attackers.

As shown in Fig. 1, the HSDirSniper attack operates on a straightforward premise. An attacker, having pre-deployed a multitude of hidden services, churns out a vast array of malicious descriptors. These are then strategically forced upon the responsible HSDirs of the target hidden service. Once the memory burden of stored descriptors overshadows the stipulated threshold ($Q$ in Eq. 3), the responsible HSDir invokes its aging mechanism. It systematically purges the oldest descriptors in hourly intervals until the memory footprint shrinks below $0.2 \times Q$. The crux of the HSDirSniper attack lies in inundating the target HSDir with a torrent of descriptors within a single hour. This ensures that the HSDir is coerced into expunging all its stored descriptors—encompassing both the malevolent ones from the attacker and the genuine descriptor of the target hidden service. To empower this attack's precision, we have augmented *two interfaces* of Stem [21], a controller library dedicated to manipulating the Tor process.

The two interfaces include: (1) **Obtaining the responsible HSDir of the target hidden service.** (2) **Uploading the descriptors to the specified relay**: The hidden service selects the responsible HSDirs based on Eq. 1 and Eq. 2, but we can modify this strategy by altering the source code of the hidden service. This allows us to avoid incurring significant costs to ensure that our hidden service chooses the same responsible HSDir as the target hidden service.

### 3.2 A Theoretical Framework to Evaluate the Cost of HSDirSniper

**Cost Function**. Since our attack employs malicious descriptors to fill HSDir's descriptor cache, the cost $\mathfrak{R}$ (MB) of the attack is represented by

$$
\mathfrak{R} = \sum_{i=0}^{\ell-1} Q_i, \tag{4}
$$

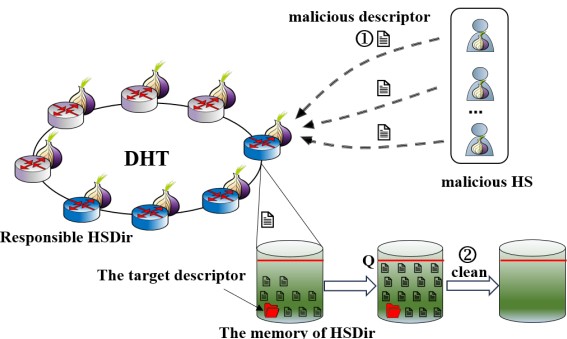

**Figure 1: The principle of HSDirSniper.**

where $\ell$ represents the number of HSDirs, $Q_i$ denotes the descriptor caching threshold $Q$ of the $i^{th}$ HSDir, as defined in Eq. 3. The challenging initial step involves the estimation of the value of $\ell$.

It is assumed that the DHT consists of $N$ (a constant) HSDirs. Each HSDir ($HSDir_i$) is assigned an index denoted as $I_i$, where $i \in [0, N-1]$ and the calculation of $I_i$ is based on Eq. 2. The interval between the $i^{th}$ HSDir and the $(i-1)^{th}$ HSDir is represented as $d_i$, with $d_i = I_i - I_{i-1}$. If a descriptor ID falls within the interval $d_i$, the $i^{th}$ HSDir is considered **responsible** and capable of receiving the descriptor. Consequently, the probability of a descriptor ID falling within the interval $d_i$ is computed by

$$
\eta_i = \frac{d_i}{\sum_{j=0}^{N-1} d_j} \tag{5}
$$

**Expected number of of responsible HSDirs**. In Section 2.1, we mentioned that a hidden service maintains both current and next descriptors and independently selects 8 responsible HSDir to upload the descriptors. Thus the responsible HSDirs for current descriptor and the responsible HSDirs for next descriptor are possibly *intersecting*. We proposes a set of theoretical calculations to derive the expected value (denoted as $\ell$) of the number of responsible HSDirs for $n$ hidden services. For the sake of facilitating the comprehension of the derivation process, we consider a simplified scenario where the DHT exclusively stores the descriptors of $n$ target hidden services that we are interested in. In this context, let $X$ denote the number of HSDirs responsible for storing the descriptors of the target hidden services, and $Y$ be the count of HSDirs that do not hold such descriptors. Hence,

$$
\ell = E(X) \text{ and } X = N - Y \tag{6}
$$

**Expected number of of irresponsible HSDirs**. Combining the nature of expectation and Eq. 6, $\ell = E(X) = N - E(Y)$. To solve the value of $E(Y)$, we define an event

$$
\varphi_i = \begin{cases} 1, & HSDir_i = \emptyset. \\ 0, & HSDir_i \neq \emptyset. \end{cases} \tag{7}
$$

where $HSDir_i = \emptyset$ represents $HSDir_i$ still does not store any descriptors after $n$ hidden services choose the responsible HSDirs, $HSDir_i \neq \emptyset$ means that $HSDir_i$ holds at least one descriptor. Since

**Figure 2: Dividing the $DHT_{current}$ into 3 regions.**

$Y = \sum_{i=0}^{N-1} \varphi_i$, we have

$$
\begin{aligned}
E(Y) &= E\left(\sum_{i=0}^{N-1} \varphi_i\right) = \sum_{i=0}^{N-1} E(\varphi_i) \\
&= \sum_{i=0}^{N-1} (1 \times p\{HSDir_i = \emptyset\} + 0 \times p\{HSDir_i \neq \emptyset\}) \quad (8) \\
&= \sum_{i=0}^{N-1} p\{HSDir_i = \emptyset\}
\end{aligned}
$$

**Estimating $p_i$** Let $p_i$ be the probability that the $HSDir_i$ does not store any descriptors after a certain hidden service has selected the responsible HSDirs. Since such selection is independent, Eq. 8 can be written as:

$$
E(Y) = \sum_{i=0}^{N-1} p\{HSDir_i = \emptyset\} = \sum_{i=0}^{N-1} (p_i)^n \quad (9)
$$

Note that each hidden service maintains current descriptor and next descriptor, and the selection of the responsible HSDirs for `current descriptor` and `next descriptor` is independent and identical. Therefore, we only need to consider one of current descriptor or next descriptor.

**Estimating $p_i^1$** We use $B_i$ to denote the event that the $HSDir_i$ does not store any descriptor after a hidden descriptor has selected the responsible HSDirs for `current descriptor`. Discrete random variables $A_1$ and $A_2$ indicate that the descriptor ID1 and ID2 of current descriptor fall in a range of the $DHT_{current}$, respectively. Let $p_i^1$ represents the probability of event $B_i$ transpiring. Therefore, we have $p_i^1 = P(B_i)$ and $p_i = (p_i^1)^2 = (p(B_i))^2$. As shown in Fig. 2, we divide the $DHT_{current}$ into 3 regions in order to solve for $p_i^1$, where $a_1$ and $a_2$ are the ranges of $[I_{i-3}, I_i]$ and $[I_{i-7}, I_{i-4}]$ in which a descriptor ID falls, respectively, and $a_3$ indicates that the descriptor ID falls in a location other than $a_1$ and $a_2$ (i.e., $ID \notin [I_{i-7}, I_i]$). According to Total Probability Theorem, we have

$$
\begin{aligned}
p_i^1 = p(B_i) = \ &p(A_1 = a_1) \times p(B_i|A_1 = a_1) \\
&+ p(A_1 = a_2) \times p(B_i|A_1 = a_2) \quad (10) \\
&+ p(A_1 = a_3) \times p(B_i|A_1 = a_3)
\end{aligned}
$$

**Three cases** Then we analyse the three components of Eq. 10 in depth.

First, when $A_1 = a_1$, we have $desc\_id1_{current} \in [I_{i-3}, I_i]$, at which point $HSDir_i$ is able to collect and store the descriptor. Thus, $p(B_i|A_1 = a_1) = 0$.

Second, when $A_1 = a_2$, we use Total Probability Theorem:

$$
\begin{aligned}
p(B_i|A_1 = a_2) = \ &p(A_2 = a_1) \times p(B_i|A_1 = a_2, A_2 = a_1) \\
&+ p(A_2 = a_2) \times p(B_i|A_1 = a_2, A_2 = a_2) \quad (11) \\
&+ p(A_2 = a_3) \times p(B_i|A_1 = a_2, A_2 = a_3)
\end{aligned}
$$

For the first part of Eq. 11, since $HSDir_i$ is able to collect and store the descriptor in the case $A_2 = a_1$, we have $p(B_i|A_1 = a_2, A_2 = a_1) = 0$. And then we analyse the second part of Eq. 11, $A_1 = a_2$ and $A_2 = a_2$ indicates that when $desc\_id1_{current}$ and $desc\_id2_{current}$ of current descriptor both select $a_2$, the final $desc\_id2_{current}$ will fall into the $a_1$ region due to the presence of a skip mechanism (See Section 8.3) in the selection of responsible HSDirs, and therefore $p(B_i|A_1 = a_2, A_2 = a_2) = 0$. Finally, we consider the third part of Eq. 11. Since $A_1 = a_2$ and $A_2 = a_3$, $desc\_id1_{current} \in [I_{i-7}, I_{i-4}]$ and $desc\_id2_{current} \notin [I_{i-7}, I_i]$, $HSDir_i$ fails to collect the descriptor, so $p(B_i|A_1 = a_2, A_2 = a_3) = 1$. In summary, it can be deduced that $p(B_i|A_1 = a_2) = p(A_2 = a_3)$.

Third, using the same analysis method of $p(B_i|A_1 = a_2)$ for $p(B_i|A_1 = a_3)$, we have $p(B_i|A_1 = a_3) = p(A_2 = a_2) + p(A_2 = a_3)$. In the end, we simplify Eq. 10 as

$$
\begin{aligned}
p_i^1 = \ &p(A1 = a_2) \times p(A_2 = a_3) \\
&+ p(A1 = a_3) \times (p(A_2 = a_2) + p(A_2 = a_3))
\end{aligned} \quad (12)
$$

Combining Eq. 5 with Eq. 12, thus (See Section 8.5 for details)

$$
p_i^1 = \left(1 - \sum_{j=i-7}^{i} \eta_j\right) \times \left(1 - \sum_{j=i-7}^{i} \eta_j + 2 \times \sum_{j=i-7}^{i-4} \eta_j\right) \quad (13)
$$

**Counting the distribution of indexed distances**. To make the results concise, it is necessary to count the distribution of indexed distances for each HSDir in the DHT. From June 6, 2023, to July 6, 2023, we conducted an examination of the indexed distances of HSDirs within the DHT, as depicted in Fig. 3. While the observed distances exhibited significant variations, it is noteworthy that the average distance remained remarkably consistent. Consequently, it becomes a reasonable proposition to employ the average spacing values as a means to streamline the simplification of Eq. 5. We let $\eta_i = \frac{1}{N}$, where $N$ denotes the length of the DHT.

Finally, we conclude the expected value (denoted as $\ell$) of the number of responsible HSDirs for $n$ hidden services in a DHT is

$$
\ell = N - \sum_{i=0}^{N-1} (p_i^1)^{2n} = N - N \times \left(\frac{N-8}{N}\right)^{2n} \quad (14)
$$

Given that the size of the descriptors created by the malicious hidden service is consistently pegged at $C$, we can deduce the requisite number of descriptors to instigate the HSDir's descriptor cache cleanup. It is calculated by

$$
n = Q/C, \quad (15)
$$

where $n$ represents the number of our descriptors, the descriptor caching threshold $Q$ is defined in Eq. 3.

By substituting Eq. 14 and Eq. 15 into Eq. 4, we derive the cost of the HSDirSniper attack:

$$
\Re = \sum_{i=0}^{\ell-1} Q_i = \sum_{i=0}^{\ell-1} n_i \times C \quad (16)
$$

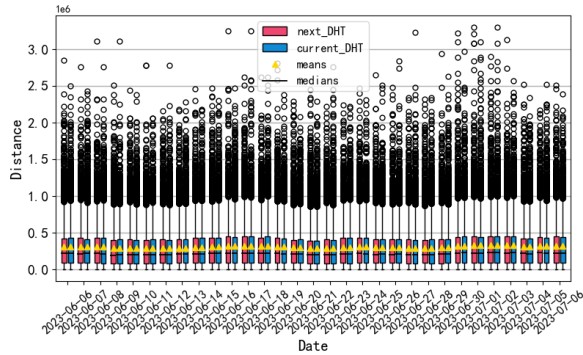

**Figure 3: Distribution of indexed distances for each HSDir in the DHT.**

## 3.3 Effectiveness of HSDirSniper

The control ratio, represented by the variable $S$, is introduced as a metric to quantify the effectiveness of a single successful attack on the targeted hidden service, which is calculated by

$$S = \chi/R, \tag{17}$$

where $\chi$ indicates the attack duration required to execute a single HSDirSniper attack, $R$ is the time interval between the present upload of the descriptor and its previous upload. For instance, when $S = 90\%$, it signifies that the target hidden service remains unavailable for 90% of the specified time period.

## 4 EXPERIMENTS

In this section, we delve into a comprehensive examination of the HSDir attack cost. Subsequently, we conduct a meticulous measurement of the control ratio within the context of launching an HSDirSniper attack against a hidden service. Finally, we undertake a comparative analysis between our attack and previous DoS attacks against the hidden service itself.

### 4.1 Evaluating the cost of HSDirSniper

Estimating the number of responsible HSDirs and determining the requisite number of descriptors essential for launching an HSDirSniper attack across various memory settings play a pivotal role in the calculation of the HSDirSniper attack cost.

*4.1.1 Validating the correctness of the theoretical framework.* In Section 3.2, we conducted an estimation of the number of responsible HSDirs using Eq. 14. To validate the robustness and generality of our theoretical framework, we initiated a two-fold validation process. Firstly, we employed self-generated hidden services to validate the correctness of Eq. 14. The set of randomly generated hidden services was divided into 15 groups, each comprising a different number of hidden services ranging from 10 to 150. Subsequently, we conducted 10 independent replications of the experiment for each group. In addition, we randomly selected a day's consensus file (e.g. 2023-06-10 12:00:00) to construct the DHT, where the length of DHT was 4003. The results, as depicted in Fig. 4a, reveal that the experimental values fluctuated above and below the theoretical values with an average error of ±1.86%.

Secondly, we conducted an empirical study using 100 hidden services of a phishing site within the real Tor Network. To ensure the reliability of our results, we instituted a monitoring period spanning from June 6, 2023, to June 13, 2023, and constructed the Distributed Hash Table (DHT) utilizing the daily 12:00 consensus file. The empirical findings, as depicted in Fig. 4b, reveal a notable concurrence between the experimental and theoretical values, with a mere average error of ±1.48%.

The above experiments verify that our proposed theoretical framework is correct. Fig. 4c visually represents the trends associated with both the theoretically derived value and the upper limit value, denoted as 16n, which serves as a simplistic means for estimating the number of responsible HSDirs corresponding to n hidden services. As the quantity of targeted hidden services increases, a noteworthy disparity between the theoretically computed value and the 16n becomes increasingly pronounced. Therefore, employing this theoretical framework to estimate the number of responsible HSDirs can assist attackers in more effectively allocating their attack resources.

*4.1.2 Measuring the cost of attacking HSDir with different memory configurations.* Assuming that the memory of the HSDir is known, the number of descriptors required to launch a single attack on it can be calculated from Eq. 15. Next we need to evaluate its correctness. Within the real Tor Network, we implant HSDirs equipped with distinct memory configurations, specifically 2GB, 4GB, 6GB, 8GB, 16GB, and 24GB. Subsequently, we proceed to fill their descriptor caches with malicious descriptors, each of which is 86,550 bytes[1] in size. Our objective is to record the number of malicious descriptors necessary to deplete the HSDir's descriptor cache. Fig. 5a presents the outcomes stemming from Eq. 15 and the experimental values. It becomes evident that the number of malicious descriptors introduced by the attacker in reality is less than the theoretical value calculated by Eq. 15. This discrepancy arises due to the concurrent function of HSDir in providing storage services to normal hidden services while the attack is in progress. Consequently, the attacker is only required to fill the remaining portion of the descriptor cache. Given the degree of congruence between the theoretical values and experimental values, characterized by an error margin of less than 1.5%, we are justified in utilizing the theoretical value to calculate the cost of HSDirSniper attack.

By utilizing Eq. 14 and Eq. 15, we can assess the cost of an HSDirSniper attack. As an illustrative example, for a hidden service whose responsible HSDirs have an average memory of 8GB, the HSDirSniper attack requires the cost of $8 \times 0.75 \times 16 \times 1024 = 98304 MB$ of malicious descriptors. This would require $98304 \times 1024 \times 1024/86550 = 1190979$ malicious descriptors.

### 4.2 Evaluating the effectiveness of HSDirSniper

In the **real Tor Network**, we aim to measure the attack duration $\chi$ (Eq. 17), capture the statistical propensity $R$ (Eq. 17) for descriptor re-uploading, and ascertain the control ratio $S$ (Eq. 17).

*4.2.1 Experiment Setup.* Our assessment was rooted in a real-world scenario on the Tor Network. Adhering to ethical guidelines, we set up a decoy hidden service with a bandwidth of 1 Gbit/s to act as

---

[1]See Section 8.2

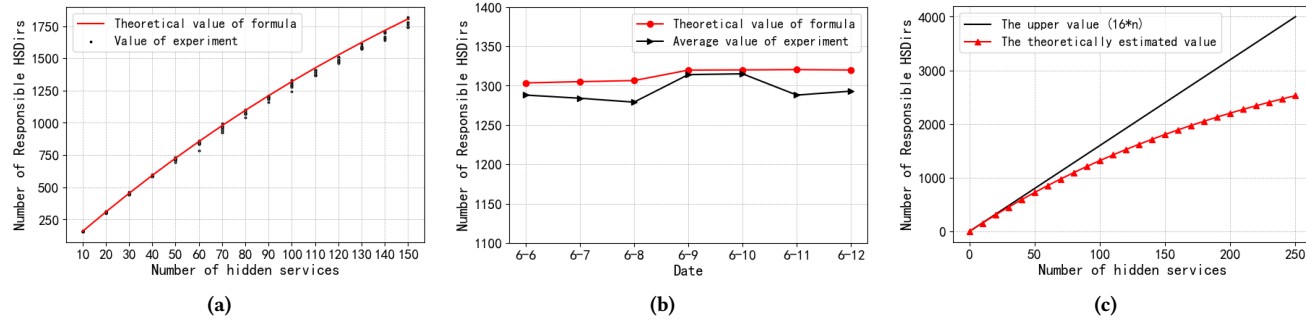

(a)                                                        (b)                                                        (c)

Figure 4: (a) Self-generated hidden services to validate theoretical framework; (b) Validating the theoretical framework using 100 real hidden services; (c) Estimated and upper limit values for the number of Resbonsible HSDirs.

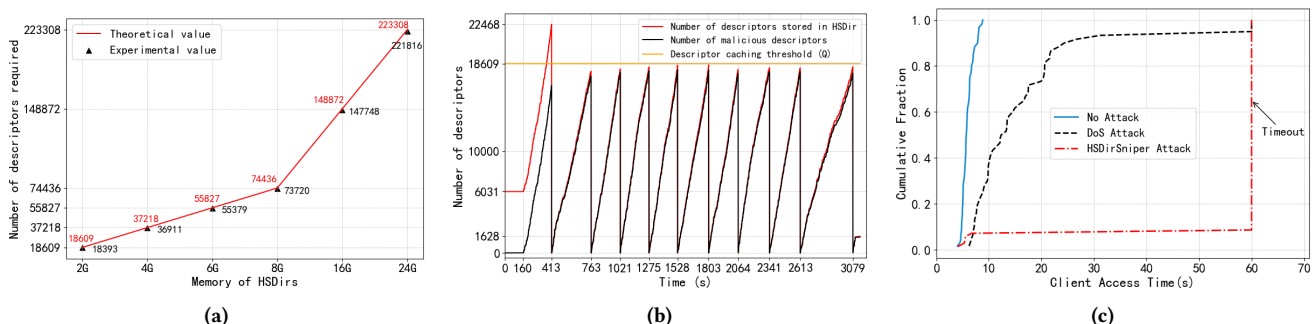

(a)                                                        (b)                                                        (c)

Figure 5: (a) Relationship between the number of malicious descriptors and the memory of target HSDir; (b) Launching HS-DirSniper 10 times against a 2G RAM HSDir; (c) Time distribution of client access to the target HS.

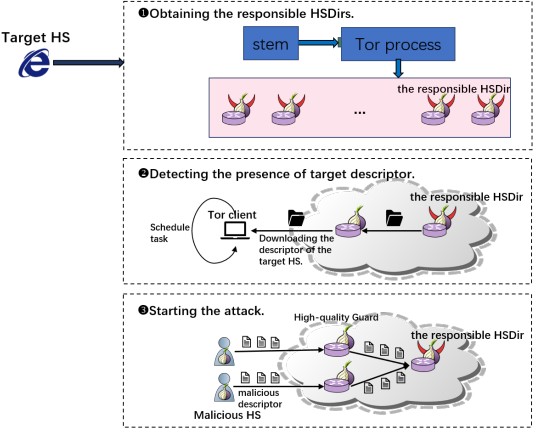

Figure 6: HSDirSniper attack system.

the victim. By adjusting the source code, we channeled descriptors to sixteen of our relays, each equipped with 2GB RAM . It's worth noting that these relays operated on Tor version 0.4.7.13[2]. However, we integrated specific modifications, allowing the capture of crucial descriptor metrics such as *ID*, *size*, and *volume*. This tailored setup provided clarity in segregating malicious descriptors and yielded insights into the relationship between descriptor volume and the relay's memory capacity.

For our malicious hidden services, we used the version 0.4.4.6, to generate and dispatch a substantial volume of malicious descriptors. Fig. 6 visually captures the experiment. Our initial step involved pinpointing the responsible HSDirs of the victim hidden service for HSDirSniper attack, which in this case were the 16 relays we had pre-configured. If the descriptors of the target hidden service exists, the malicious hidden services commenced the transmission of descriptors to these HSDirs continuously until the descriptor is cleared; otherwise, the attack was halted.

*4.2.2 Assessing the Control Ratio of HSDirSniper.* **Attack Duration.** In Section 8.2, we introduced six optimization techniques designed to expedite the attack process. Of these, three optimizations— selecting superior guard relays, adopting circuit multiplexing, and modulating the thread count for each hidden service—are paramount to assess in terms of their influence on the attack duration. While certain strategies, like curtailing the circuit length, are widely recognized for their efficacy in shortening attack times, we opt not to delve into them here. To deliver a thorough assessment, the subsequent impact of these three key optimization methods on the attack duration is presented in Table 1.

In Table 1, the column labeled #threads denotes the concurrent hidden services initiated by a malicious hidden server. For instance, 50 threads means that the server spawns 50 hidden services and simultaneously uploads 2×50 descriptors, encompassing both "current descriptor" and "next descriptor". Our results suggest that selecting an appropriate value of #threads, such as setting it

---

[2]This is the latest version when we finished our manuscript.

**Table 1: Attack time ($s$) of three optimization schemes on .**

| #threads | w/o high-quality guard | w/o circuit multiplexing | All optimizations |
|---|---|---|---|
| 1 | 0.597 | 1.370 | 0.116 |
| 50 | 6.014 | 6.048 | 4.154 |
| 100 | 13.999 | 22.282 | 7.253 |
| 200 | 37.449 | 82.174 | 33.930 |
| 500 | 68.827 | $\infty$ | 61.430 |
| 1000 | 204.301 | $\infty$ | 133.643 |

to 100, enhances the efficiency of resource utilization on the hidden server, thereby minimizing the attack duration. Moreover, the role of "`high-quality guard`" and "`circuit multiplexing`" in reducing the time spent on the attack are notably significant.

Based on the experimental setting in Section 4.2.1 and above optimization techniques, we initiated 5 malicious hidden services maintaining 60MB/s bandwidth to upload descriptors, and staged 10 attacks against our own victim hidden service. Fig. 5b illustrates an instance in which one of the responsible HSDirs was subjected to the attack. Before launching the assault, the HSDir had already accumulated 6,031 descriptors. By the 160-th second, our offensive commenced. Given the pre-existing descriptors within the HSDir, we didn't require the full 18,609 (the theoretical value) malicious descriptors to compel the HSDir to purge all of its descriptors. Instead, only 16,929 were needed. Additionally, as our attack was simultaneous with the HSDir's regular descriptor reception, the actual count of malicious descriptors used in subsequent attacks never exceeded the theoretical prediction. In the end, our results showed that, on average, the HSDirSniperattack against a hidden service whose responsible HSDirS equipped with 2GB RAM took roughly $\chi = 4.865$ minutes.

**Statistical Propensity.** Subsequently, we sought to derive the statistical value, denoted as $R$, characterizing the interval for descriptor re-uploads. Analyzing the descriptor upload timings captured by our strategically positioned HSDirs between January 1, 2023, and July 1, 2023, we observed a distinct pattern. As Fig. 8b illustrates, a significant portion of the descriptor re-upload intervals cluster within the 60 to 120-minute bracket. This behavior is consistent with the recommended settings in the Tor protocol for descriptor re-upload (as mentioned Section 2.1). In addition, the presence of intervals spanning between 1 to 60 minutes could signify either transient instability of the associated IPOs or the hidden services refreshing their consensus file. In a quest to offer a more nuanced perspective on the descriptor re-upload interval, we determined its median, which stands at $R = 68.9$ minutes.

**Control Ratio Estimation.** To gauge the potency of our attack, we infused the obtained values of $\chi$ and $R$ into Eq. 17, resulting in a control ratio estimate of $S = 92.9\%$. This implies that during the attack's duration, the targeted hidden service is inaccessible approximately 92.9% of the time. It's imperative to highlight that this estimation is based on an attack targeting an HSDir with a memory capacity of 2G. We postulate that for HSDirs boasting a memory greater than 2G, the attack duration can be further truncated by simply increasing the upstream bandwidth of malicious hidden services.

## 4.3 Comparison of methods

We have undertaken a comparative analysis using a method that directly targets the hidden service itself, referred to as `DoS Attack`. This DoS Attack is predicated on the principle of employing a Tor client to download substantial quantities of the application layer content from the hidden service[12], thereby causing a consumption of the hidden service's bandwidth. In order to ensure equitable evaluation, we have constrained the attack bandwidth for both the `DoS Attack` and our `HSDirSniper Attack` to 60MB/s. Subsequently, we have observed the time required for a standard client to access the target hidden service under three different conditions over a 1-hour period: `No Attack`, `DoS Attack`, and `HSDirSniper Attack`, respectively. Notably, the target hidden service continues to operate under the same experimental setup as outlined in Section 4.2.1 and the client access timeout is set to 60 seconds.

The experimental results, as depicted in Fig. 5c, establish a baseline by measuring the time taken for a client to access the target hidden service in the absence of any attack, yielding an average value of 5.1 seconds. When subjected to the pressure of the `DoS Attack`, the average client access time significantly increases to 17.2 seconds, indicating an extension of 12.1 seconds in access time. In contrast, while the `HSDirSniper attack` is ongoing, the client maintains access times consistent with the baseline. However, following the completion of the attack, the client remains in an access timeout state. It is obvious that under identical attack bandwidth conditions, `HSDirSniper Attack` demonstrates greater efficacy compared to the `DoS Attack`, which directly targets the hidden service itself.

## 5 DISCUSSION

### 5.1 Mitigation of HSDirSniper attack

In this section, we focus on addressing the shortcomings of HSDir to implement mitigation of the HSDirSniper attack.

**The first step.** Developing a mechanism for HSDir to detect whether a descriptor should be received or not. According to the hidden service protocol, a hidden service descriptor has at most 16 replicas, and each replica selects at most 128 consecutive HSDir. Due to the skipping mechanism used in selecting the responsible HSDir, the maximum deviation between the first and the last responsible HSDir is $D = 16 \times 128$. It is noteworthy that the deviation range is too large so that can be easily exploited by an attacker. Through empirical observation of the Tor Network spanning an extended temporal period, it has been discerned that the default parameters, specifically the number of replicas and the quantity of consecutive HSDirs, adequately suffice in maintaining the high availability of descriptors. Therefore, we suggest that the maximum deviation is taken as $D = 2 \times 4 = 8$. Upon receiving a descriptor, the $HSDir_i$ calculates the deviation in the DHT between itself and the first responsible HSDir for that descriptor. If the deviation is greater than $D$, the descriptor is not accepted.

**The second step.** Removing descriptors for undesirable hidden services instead of the oldest ones. Specifically, HSDir maintains an internal hash table to evaluate each stored descriptor. For each legal descriptor, the update time and the number of times the descriptor has been downloaded are recorded. We consider a normal descriptor to have a regular update time (typically 60-120 minutes)

and a certain number of clients using it. On the contrary, if a descriptor is frequently uploaded or never used after upload, then HSDir should not provide services for such a descriptor.

## 5.2 Limitations and Ethics

While the HSDirSniper attack is capable of blocking arbitrary hidden services, it undeniably poses a significant threat to non-targeted hidden services as well. This is due to the fact that the descriptors cleared by HSDir include both the descriptors of the target hidden service and those of the non-targeted ones. In this paper, we refrain from attacking uncontrolled HSDirs as it would adversely affect the communication of legitimate users within the Tor Network. We have reported these vulnerabilities to the Tor Project.

## 6 RELATED WORK

### 6.1 DoS Attacks against Tor Hidden Services

DoS attacks aimed at Tor *hidden services* (HS) have emerged as a critical challenge in preserving the reliability and privacy of the Tor Network [3, 10]. These attacks can be categorized into distinct areas based on the specific component of the hidden service infrastructure they target.

**Attacking the HS itself.** Within the Tor protocol's design, there exists an inherent asymmetry: a malicious client can send a relatively small message to the HS, which in turn is compelled to perform substantial computational work in response. This imbalance can be exploited to initiate DoS attacks against the HS. Rochet and Pereira [15] demonstrated this vulnerability by using a rogue client to flood the HS with an overwhelming number of requests. This forces the HS to establish multiple HS-RP (Hidden Service - Rendezvous Point) circuits, thereby placing considerable stress on the network resources of the HS host.

**Attacking the guard relay of HS.** The guard relay acts as the pivotal entry point for a HS accessing the Tor Network. By focusing their efforts on the guard relay, attackers can disrupt the hidden service's stable connection with the Tor Network. Prior studies [1, 9, 11, 12] have used methods to exhaust the CPU, memory, and bandwidth of the guard relay. As a result, the relay's capability to seamlessly relay traffic for the HS is significantly hampered.

**Attacking the responsible HSDir of HS.** The responsible HSDir is integral to the Tor hidden service infrastructure as it stores the descriptors (contact details of the HS). This pivotal role also marks it as a vulnerable point of attack. In the investigations conducted by [2, 16], it has been shown that malevolent entities can deploy rogue relays to masquerade as the legitimate responsible HSDirs for a specific HS. Such impersonation disrupts the natural flow of communication, effectively barring clients from connecting with their intended HS.

### 6.2 Countermeasures for DoS Attacks on HS

To increase prevalence of DoS attacks on hidden services, both the Tor Project and the research community have devised a range of countermeasures to bolster the resilience of the Tor Network.

**Countermeasures for attacks on HS.** Döpmann et al. [7] proposed *Onion Pass*, a novel extension to the Tor protocol. This allows clients to validate their authenticity via cryptographic tokens. Consequently, onion services can distinguish and prioritize legitimate

users over unauthenticated ones, ensuring sustained availability amidst a barrage of spurious requests. Fraser et al. [8] advocate for a Proof-Of-Work system, which narrows the computational disparity between the service and its potential attackers. By requiring clients to solve a Proof-of-Work puzzle as a precursor to access, it becomes infeasible for adversaries to inundate the service. Further enhancing the robustness of HS, the Tor Project unveiled *Onionbalance* [20]. This tool facilitates the distribution of Tor onion service requests across multiple backend Tor instances. Beyond load-balancing, *Onionbalance* enhances the resilience and reliability of onion services by eliminating potential single points-of-failures.

**Countermeasures for attacks on the Guard Relay.** Addressing the vulnerabilities of relay resource exhaustion, the Tor Project refined its system's multi-threaded architecture, thus amplifying Tor's ability to manage an augmented volume of data cells. As an additional safeguard, the Tor Project integrated an adaptive out-of-memory circuit killer within Tor [19]. This feature is only invoked when memory resources dwindle, targeting the circuit possessing the eldest front-most cell in its queue.

**Countermeasures for attacks on the Responsible HSDir.** In light of the vulnerabilities identified in V2 HS concerning the HSDir, the Tor Project rolled out V3 HS [23]. This iteration incorporates randomized vectors when constructing the distributed hash table (DHT), obfuscating the responsible HSDirs for a given HS. This strategic change effectively counters the attack vectors highlighted by Tan et al. [16] and Biryukov et al. [2].

In the context of high confrontation, we propose a unique attack that focuses on forcing the responsible HSDirs to clean up the descriptors of the target hidden services. In contrast to DoS attacks that directly target hidden services, our attack exhibits more efficacy and is not perceived by hidden services. Notably, our attack does not necessitate an extensive effort to ascertain the identities of guard nodes, as compared to attacks that target the guards of the hidden services, which makes our attack more versatile. Moreover, when juxtaposed with the Eclipse attack previously conducted by Tan et al. [16], our method demonstrates a superior capability to accurately track the responsible HSDirs, rendering it a more pragmatic and adaptable attack within the context of real Tor Network.

## 7 CONCLUSION

In this paper, we introduce a new attack that forces the responsible HSDir to clear the descriptor and thus block arbitrary hidden services. We conduct comprehensive experiments on a real Tor Network and reveal that an adversary wielding a 60MB/s attack bandwidth can render a hidden service inaccessible for approximately 92.9% of its operational duration, when the responsible HSDirs are equipped with a 2GB RAM capacity. Furthermore, we put forth a general and dependable theoretical framework to estimate the number of responsible HSDirs, addressing a notable lacuna within the current body of research in this domain. In conclusion, we engage in a discourse pertaining to prospective solutions aimed at mitigating HSDirSniper attack. These solutions are envisioned for implementation in forthcoming research endeavors.

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

# 8 APPENDICES

## 8.1 Components of the Hidden Service Mechanism

The hidden service mechanism leverages Onion addresses in lieu of traditional IP addresses. This unique approach ensures enhanced anonymity for server-side applications, including web services and instant messaging platforms. Broadly, the hidden service mechanism is composed of five pivotal components: Hidden Service, Introduction Point, Rendezvous Point, Hidden Service Directory and Tor Client.

**Hidden Service.** Acting as a localized proxy, the hidden service facilitates server-side applications' integration with the Tor Network. It produces an Onion address—a 62-byte string terminating with the '.onion' extension—which stands as an anonymized replacement for the server's actual IP address. Diverging from conventional domain name resolutions, the hidden service adopts a unique mechanism. It first utilizes a descriptor, essentially a text file containing both plaintext and encrypted segments, to capture its contact details. Following specific rules, the hidden service then earmarks multiple HSDirs within the DHT to act as the responsible HSDirs. Their role? To host and propagate the freshly-minted descriptor. For any client to establish a connection with the hidden service, it becomes imperative to first identify the relevant HSDir associated with the desired service and subsequently procure its descriptor. As such, the descriptor emerges as a linchpin, central to the hidden service's ability to offer its services to external clients.

**Introduction Point (IPO).** The Introduction Point, often abbreviated as IPO, is a specific Tor relay. It's selected at random by the hidden service, serving a pivotal role in channeling connection requests made by the client directly to the hidden service.

**Rendezvous Point (RPO).** Denoted as RPO, the Rendezvous Point is a Tor relay handpicked by the client. Its main function is to amalgamate connections from both the client and the hidden service, ensuring seamless data transfer between the two entities.

**Hidden Service Directory (HSDir).** These are not just any Tor relays but specialized ones, identifiable by their distinct HSDir flag. Their primary role is to archive the descriptors associated with hidden services. Recognizing the potential load on these directories, each HSDir crafts a distributed hash table (DHT), relying on a unique formula to derive its node index. Highlighting a feature of the v3 hidden service mechanism, it's worth noting that every hidden service concurrently oversees two descriptors. Each of these descriptors meticulously identifies and then uploads to a set of eight chosen HSDirs.

**Tor Client.** Acting as the user's gateway to the Tor Network, the Tor client functions as a localized proxy. It equips user-end applications, including web browsers, to interface with the Tor Network, effectively translating standard user traffic into the specialized Tor protocol format.

Fig. 7 showcases the sequence of operations required to form a connection between the Tor client and the designated hidden service. It's imperative to note that the hidden service juggles both the "current descriptor" and the "next descriptor" due to the intrinsic delay in client consensus files. This ensures a seamless connection for clients equipped with either the newer or older consensus. The following steps elucidate the process:

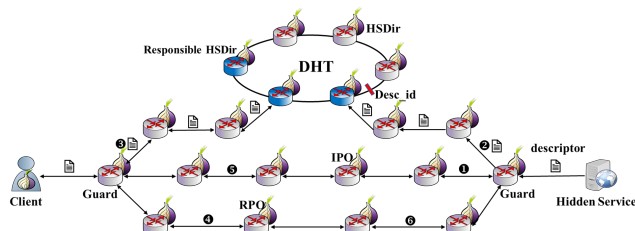

**Figure 7: Components of the hidden service mechanism and the client access process.**

- ① Descriptor Selection: Each descriptor elects three relays as IPOs. A 3-hop circuit to each IPO is subsequently established.
- ② Information Update: The hidden service logs the IPO information into the appropriate descriptor. Eight responsible HSDirs on the DHT are chosen for each descriptor, to which the descriptors are then uploaded.
- ③ Client Access: When a Tor client aims to access a target hidden service, it decides to use either the current or next descriptor. This decision is influenced by the temporal data (specifically, the "valid-after" field) in its consensus file. Using the target hidden service's onion address, the client computes the location of responsible HSDirs on the DHT and fetches the hidden service descriptor.
- ④ RPO Selection: The Tor client opts for a Tor relay to act as the RPO. A 3-hop circuit to the RPO is established, and a rendezvous cookie is generated for hidden service authentication.
- ⑤ IPO Connection: Post decryption of the descriptor, the client obtains three IPOs. One IPO is chosen at random for a 4-hop circuit connection. The client sends a request, inclusive of the RPO details and rendezvous cookie, to the hidden service via the selected IPO.
- ⑥ RPO Circuit Establishment: On receipt of the request, the hidden service forms a 4-hop circuit to the RPO. Using the rendezvous cookie, authentication with the RPO is completed.
- ⑦ Circuit Linking: The RPO connects the two circuits originating from both the client and the hidden service. Finally, the Tor client and hidden service communicate via the resulting 6-hop circuit.

## 8.2 Reducing attack time

To maximize the effectiveness of the HSDirSniper attack, it is imperative to minimize the duration required for the attacker to upload descriptors. Eq. 18 represents the relationship between the time $T$ required to attack a responsible HSDir and the per-descriptor upload time $T_i$. There are three primary factors that limit the parameter $T_i$: the time spent on descriptor construction $T_{desc,i}$, the time spent about the circuit $T_{circ,i}$, and the performance of both the attacker's malicious hidden services and the target HSDir $T_{machine,i}$. In this paper, six optimisation solutions are proposed to reduce the time of attack ($T$).

$$T = \sum_{i=1}^{n} T_i = \sum_{i=1}^{n} (T_{desc,i} + T_{circ,i} + T_{machine,i}) \quad (18)$$

(1) Reducing the number of descriptors ($n$).

① Expanding the capacity of each descriptor. In Section 2.2, we mentioned that the size of one descriptor is limited to 50,000 bytes. According to the strategy for storing descriptors in HSDir, the ciphertext section of the descriptor is stored twice. We expanded the ciphertext section of the descriptor with additional bytes, and finally, the malicious descriptor we constructed occupies around 86,550 bytes in HSDir, which is 3.5 times larger than a normal descriptor.

(2) Reducing the time spent on buliding descriptors ($T_{desc,i}$).

② Skipping the step of selecting the IPOs in the process of building the descriptor. Establishing a HS-IP circuit is a very time-consuming process. Skipping the selection step of the IPOs removes the time spent not only on establishing the circuit, but also on selecting the IPOs.

(3) Reducing the time spent on circuit ($T_{circ,i}$)

③ Reducing the length of the HS-HSDir circuit. We have reduced the HS-HSDir circuit to 1 hop from the default 4 hops, which effectively decreases the transmission delay of the circuit, i.e. HS->Guard->HSDir.

④ Choosing Guard relays with excellent transmission performance. By default, HS employs a bandwidth weighting algorithm to select Guards. However, it's important to note that Guards with high bandwidths do not necessarily guarantee consistent transmission performance. In this paper, we propose measuring the transmission performance of Guards in real time by utilizing the circuit build time as a metric.

⑤ Multiplexing the HS-HSDir circuit. The hidden service does not multiplex the HS-HSDir circuit, resulting in the creation of a new circuit every time the hidden service uploads to the same HSDir. This mechanism increases the time required for the hidden service to establish the circuit.

(4) Evaluating the data handling capabilities of malicious hidden services and HSDir ($T_{machine,i}$).

⑥ To increase the attack speed and maximize the utilization of machine resources, an attacker can deploy multiple malicious hidden services and utilize a multi-threaded design in each hidden service's Tor instance. However, HSDir has a performance limit and will discard excess data when it cannot handle it. Therefore, it is crucial to deploy a reasonable number of hidden services and carefully explore the appropriate number of multi-threads.

## 8.3 Responsible HSDir skipping mechanism

The HSDir selection for the "current descriptor" operates independently from the "next descriptor". This autonomy implies potential overlaps between the eight HSDirs chosen by the "current descriptor" and those selected by the "next descriptor". Furthermore, for any descriptor (be it current or next), if a HSDir is pre-selected, the hidden service bypasses it in favor of the subsequent HSDir. Upon finalizing the selection of the responsible HSDirs, the descriptor is uploaded.

## 8.4 Experimental Supplementary Figure

In this section, we show some supplementary figure of the experimental procedure. Fig. 8a show the process of selecting the responsible HSDir by the hidden service, while Fig. 8b indicates the re-uploading intervals of descriptors in the real Tor Network.

## 8.5 Details of the theoretical framework derivation

In Section 3.2, we use Total Probability Theorem for $p(B_i|A_1 = a_2)$, and have $p(B_i|A_1 = a_2) = p(A_2 = a_3)$.

Using the same method to expand the full probability formula for $p(B_i|A_1 = a_3)$. we can have

$$
\begin{aligned}
p(B_i|A_1 = a_2) = \; & p(A_2 = a_1) \times p(B_i|A_1 = a_3, A_2 = a_1) \\
& + p(A_2 = a_2) \times p(B_i|A_1 = a_3, A_2 = a_2) \\
& + p(A_2 = a_3) \times p(B_i|A_1 = a_3, A_2 = a_3)
\end{aligned}
\tag{19}
$$

For the first part of Eq. 19, since $HSDir_i$ is able to collect and store the descriptor in the case $A_2 = a_1$, we have $p(B_i|A_1 = a_3, A_2 = a_1) = 0$. And then we analyse the second part of Eq. 19, when $A_1 = a_3$ and $A_2 = a_2$, $HSDir_i$ fails to collect the descriptor, and therefore $p(B_i|A_1 = a_3, A_2 = a_2) = 1$. Finally, we consider the third part of Eq. 19. Since In this case that $A_1 = a_3$ and $A_2 = a_3$, $HSDir_i$ fails to collect the descriptor, so $p(B_i|A_1 = a_3, A_2 = a_3) = 1$. In summary, it can be deduced that $p(B_i|A_1 = a_3) = p(A_2 = a_2) + p(A_2 = a_3)$. In summary, we derive Eq. 12.

From the definition of Eq. 5, $\eta_i$ is the probability that $HSDir_i$ captures the descriptor. Therefore, we have $\sum_{i=0}^{N-1} \eta_i = 1$. Finally, combining Eq. 5 with Eq. 12, we have

$$
\begin{aligned}
p_i^1 = \; & p(A_1 = a_1) \times p(B_i|A_1 = a_1) \\
& + p(A_1 = a_2) \times p(B_i|A_1 = a_2) \\
& + p(A_1 = a_3) \times p(B_i|A_1 = a_3) \\
= \; & p(A1 = a_2) \times p(A_2 = a_3) \\
& + p(A1 = a_3) \times (p(A_2 = a_2) + p(A_2 = a_3)) \\
= \; & \sum_{j=i-7}^{i-4} \eta_j \times (1 - \sum_{j=i-7}^{i} \eta_j) \\
& + (1 - \sum_{j=i-7}^{i} \eta_j) \times (\sum_{j=i-7}^{i-4} \eta_j + 1 - \sum_{j=i-7}^{i} \eta_j) \\
= \; & (1 - \sum_{j=i-7}^{i} \eta_j) \times (1 - \sum_{j=i-7}^{i} \eta_j + 2 \times \sum_{j=i-7}^{i-4} \eta_j)
\end{aligned}
\tag{20}
$$

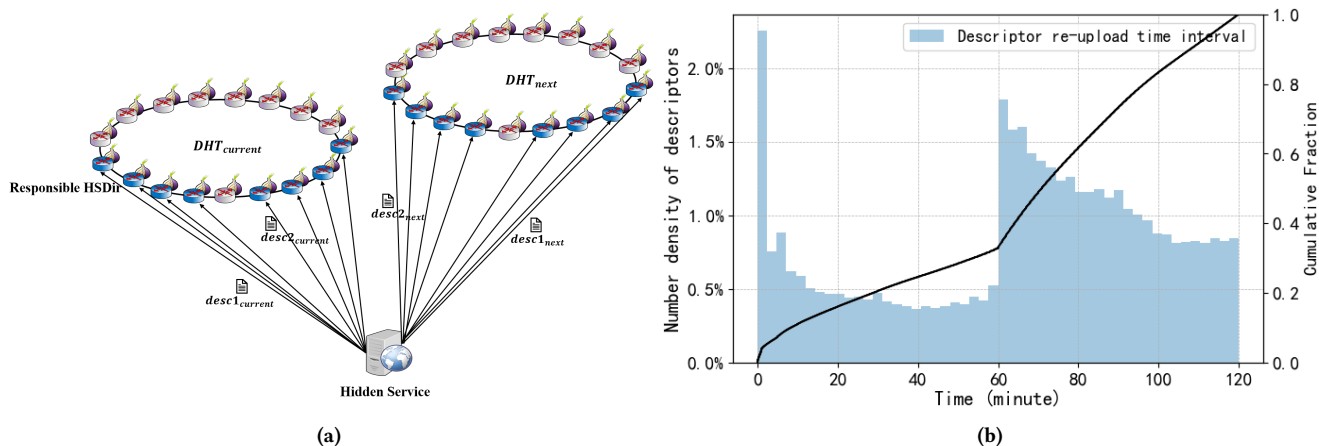

(a)

(b)

Figure 8: (a) Hidden service upload descriptors to the DHT; (b)Stats descriptor re-upload interval.

