# OpenReview forum: "HSDirSniper: A New Attack Exploiting Vulnerabilities in Tor's Hidden Service Directories"
_ACM.org/TheWebConf/2024/Conference — TheWebConf24_

### Official Review · Reviewer_GEr7 · 2023-11-17

**Novelty:** 5
**Technical Quality:** 4

**Review:**

This paper introduces a new DoS attack on Tor onion services called HSDirSniper. Instead of flooding onion services themselves with attack traffic (like in a more traditional DoS setting), the attack aims at purging the descriptor entries of a given onion service from the cache of the hidden service directories responsible for providing this information to clients. In this way, Tor clients would be unable to connect to the desired onion service, while an onion service operator would not be able to detect any attack traffic directly targetting their onion service(s).
The experiments conducted in the paper suggest that an adversary able to generate 5 dummy onion services and send ~17k descriptors towards a 2GB RAM hidden service directory within a ~5min timeframe (upload link of 60MB/s) can successfully exhaust the directory's cache and prevent clients from accessing the onion service around 92% of the time the attack in ongoing.



I found the approach taken in this work to be an interesting one. Indeed, the lack of validations performed by hidden service directories on the descriptors they receive (and with which they update their cache with) seems to be left open to adversaries to exploit (at least to some extent). I also enjoyed the theoretical analysis on the estimated costs of HSDirSniper and the discussion of potential defenses that, while arguably simple, could help tackle the identified issue. However, I'm left with a few concerns about the practicality and setting up of the experimental testbed for this work:

Practicality of the attack: The attack seems to require hefty resources from an adversary, while being far from 100% effective. While the evaluation shows that it would take around 5min for the considered attacker to force the purging of all of a 2GB RAM HSDir's stored descriptors, I suspect this "lag" would be even larger for HSDir's with larger amounts of RAM (unless the adversary is able to send flooding traffic faster than the 60MB/s assumed in this work). While the attack might still work given that most onion services re-upload descriptors every ~70 minutes, asking for onion services to refresh their descriptors at a higher rate (say every 3 minutes) would substantially complicate the attack, all without imposing a significant penalty on the HSDirs.


Attack practicality vs traditional DoS: In Section 4.1., it is noted that, for a hidden service whose HSDirs have an average memory of 8GB, the proposed attack would require about 98GB worth of descriptors to successfully flood its cache. I wonder how practical this is when compared to a more traditional attack that attempts to directly DoS an onion service server through some application-level requests. While this of course also depends on the configuration and workload of each onion service, it would have been nice to learn about some plausible comparisons.


Empirical setup: I was somewhat confused by the setup of the empirical study, which the manuscript claims to have been performed "using 100 hidden services of a phishing website within the real Tor network". Does this mean the attack was targeted at a specific onion service that was found to be replicated 100 times in the wild?


Responsible disclosure: Since the authors have reported the proposed attack to the Tor development team, it could have been useful to include Tor developers' outlook on the practicality, perceived dangers of the attack, and whether and how the proposed solutions align with Tor development plans.

Nits: The "hidden service" terminology is no longer current, and the manuscript should opt by referring to "onion services" instead. There are also some typos throughout the text (e.g., "lunch" vs. launch) which an editorial pass might help fixing. Finally, some of the mathematical deductions throughout the theoretical estimations of the attack's mechanisms are perhaps too "dry", lacking some more intuitive explanations to follow the practical implications.

**Questions:**

How much less expensive is the proposed attack vs. a traditional DoS attack, especially considering the case we'd like to make HDir very close (if not) 100% effective?

**Ethics Review Description:**

Ethics Review Description: The manuscript does not provide details on what measures have been taken to ensure that the HSDirs under the control of the authors don't include descriptors for other legitimate onion services while the experiments were underway. While I wouldn't expect this to cause a legitimate onion service to become unavailable during the experiment, it might have caused clients to fetch descriptors from alternative HSDirs, potentially leading to communication overheads. Additional details about the setup of the experiment could clear the question of whether this may have indeed been an issue.

**Ethics Review Flag:**

Yes

**Reviewer Confidence:**

3: The reviewer is confident but not certain that the evaluation is correct

**Scope:**

4: The work is relevant to the Web and to the track, and is of broad interest to the community

---

### Official Review · Reviewer_5TKR · 2023-11-20

**Novelty:** 5
**Technical Quality:** 5

**Review:**

- Strengths
    - A new DoS attack against hidden service that is imperceptible to the attack target.
    - The theoretical estimation has high similarity with experimental results.
- Weakness
    - The theoretical framework lacks clarity
    - The experimental settings and implementation details are Inadequate
    - Ethical considerations is not well discussed and addressed in the paper.
- Comments to Author

This paper introduces a DoS attack designed to block Tor hidden services by compelling responsible HSDirs to clear the descriptor of the target hidden service. The paper puts forth a theoretical framework for evaluating the cost of the proposed attack and validates the correctness of the theoretical cost estimation through experiments. Additionally, it discusses several mitigation strategies.

However, the main concerns of the paper can be categorized into two aspects.

Firstly, the theoretical cost estimation framework is confusing, and the results are not clearly derived, as summarized below:

- In Equation (16) delivering the final result of the estimated cache, $n_i$ is used without explanation. Previous equations use $n$ to denote the number of hidden services, and $i$ is used as the index of the HSDir to be attacked. The meaning of $n_i$ is unclear from prior information, making it challenging to understand how the final result is obtained from the proposed estimation framework.
- Equations (15) and (16) suggest a linear relation between the final result and $C$, which is the size of descriptors created by the malicious hidden service. Is $C$ configured by the target of the proposed attack or the attacker? If configured by the target, how could the attacker know its value? If configured by the attacker, why is the descriptor cache linearly related to $C$? Please clarify the definition of $C$ and justify why its value could be obtained by the attacker or why it is linearly related to $Q$.
- In Equation (10), DHT_current is divided into three regions. The basis for this grouping needs justification. Why are $a_1$ and $a_2$ set as $[I_{i-3}, I_i]$ and $[I_{i-7}, I_{i-4}]$, respectively? What are the implications/consequences of such assignments? Additionally, the author should explain why $p(B_i|A_1=a1)=0$ based only on the fact that desc_id1_current $\in [I_{i-3}, I_i]$.
- Minor issues:
    - What do "indexed distances" on Page 4 mean? Do they refer to $d_i$ in Eq.(5)? If so, there seems to be a problem with the conclusion that $\eta_i$ can be approximated by the average, given that the average distance remains consistent. Figure 3 shows uneven distribution between 1 and 3, with a higher density around 1.5 than around 2.5. In this case, $\eta_i$ will be overestimated by replacing Eq.5 with the average value $1/N$.
    - According to the definition $d_i = I_i-I_{i-1}$, the dummy variable in the denominator of Eq.(5) should be \sum_{j=1}^N.

Secondly, the experimental process is not well explained. The author illustrates the similarities between experimental and theoretical values to validate the theoretical estimation but does not clarify how the experiment results were obtained. The author only outlines how hidden services were allocated into different groups for the experiment, claiming these services are "self-generated." It is necessary to supplement a section to address the following questions:

- What are the hidden services in this evaluation, and how are they generated?
- What is the experimental environment? The paper claims that the assessment was rooted in a real-world scenario on the Tor Network. What exactly does this refer to, and how many HSDirs (i.e., $N$) were deployed in this network?
- If the attack was assessed in a real-world network, how were ethical issues addressed? Since the DoS attack will undoubtedly affect other services in that network, addressing such concerns cannot be trivialized by merely claiming to "Adhere to ethical guidelines," as asserted in the paper.

In summary, significant effort is required in both the theoretical and experimental parts of this paper.

**Questions:**

see above review comments

**Ethics Review Description:**

The attack experiment on the real Tor networks was not well-described and its ethical concern has not been discussed.

**Ethics Review Flag:**

Yes

**Reviewer Confidence:**

3: The reviewer is confident but not certain that the evaluation is correct

**Scope:**

4: The work is relevant to the Web and to the track, and is of broad interest to the community

---

### Official Review · Reviewer_j2yW · 2023-11-22

**Novelty:** 5
**Technical Quality:** 5

**Review:**

This paper presents HSDirSniper, a novel attack exploiting vulnerabilities in Tor's Hidden Service Directories (HSDir). The attack floods the HSDir's descriptor cache by leveraging a substantial volume of descriptors. The paper conducts experiments in real-world Tor Network environments and simulated settings to launch and evaluate the cost and effectiveness of the HSDirSniper attack.

[Strengths]

+ The paper addresses a significant and timely topic, presenting a practical DoS attack capable of blocking arbitrary Tor Hidden Service Directories.

+ Comprehensive evaluations of cost and effectiveness are conducted under different settings, employing a variety of metrics.

+ The logical organization enhances the paper's clarity.

[Weakness]

-	My primary concern is the absence of a quantitative comparison between HSDirSniper and previous attacks mentioned in the introduction. Please include such an evaluation to strengthen the evaluation section of the paper.

-	The paper lacks clarity on how HSDirSniper bypasses previous defense and detection methods. While countermeasures are discussed in the related work, a quantitative experimental analysis is needed to validate the effectiveness against existing countermeasures.

-	A typographical error ("$\to$``) is present in line 173 on page 2.

**Questions:**

Q1. Please include comparisons with previous attacks, considering both cost and effectiveness.

Q2. Provide experimental analysis of the effectiveness of the HSDirSniper attack against existing detection or defense methods.

Q3. Address the typographical error in Equation (1).

**Ethics Review Description:**

No issues found.

**Reviewer Confidence:**

2: The reviewer is willing to defend the evaluation, but it is likely that the reviewer did not understand parts of the paper

**Scope:**

4: The work is relevant to the Web and to the track, and is of broad interest to the community

---

### Official Review · Reviewer_JBSL · 2023-11-23

**Novelty:** 4
**Technical Quality:** 4

**Review:**

The paper introduces yet a new attack for Tor Networks. In particular, this one forces the HSDir to clear the descriptor, blocking hidden services. They conducted an evaluation of the attack in a real network, showing its effectiveness. While I believe the attack is worth to know, I find the countermeasures and discussion superficial in this paper.

**Questions:**

Which are the countermeasures of your attack?

**Reviewer Confidence:**

2: The reviewer is willing to defend the evaluation, but it is likely that the reviewer did not understand parts of the paper

**Scope:**

3: The work is somewhat relevant to the Web and to the track, and is of narrow interest to a sub-community

---

### Official Review · Reviewer_48us · 2023-11-23

**Novelty:** 6
**Technical Quality:** 5

**Review:**

This paper demonstrates a new method for denial-of-service attack on Tor hidden services. Rather than attacking the service directly, the attack poisons the _hidden service directory_, which is a distributed hash table (DHT). Each hidden service uploads its directory entry to this DHT, typically selecting 8 of the Tor nodes participating in the DHT (these nodes are referred to as "HSDirs" in the paper) to receive copies in a semi-randomized fashion. The attacker identifies the HSDirs holding the directory entry for the service they wish to force offline, and floods them with junk directory entries until it is forced to discard old entries -- including the targeted service's entry -- to avoid running out of RAM. If the attack succeeds, the targeted hidden service will become unreachable because clients cannot look it up in the directory.  The paper offers a detailed theoretical analysis of the probability that this attack will succeed, and a test on the real Tor network (against a hidden service and HSDir nodes controlled by the authors).  They conclude with two concrete suggestions for how the hidden service directory can be hardened against this attack.

Content suppression attacks on distributed hash tables are nothing new, but I'm only aware of one previous instance of such an attack _on Tor's hidden service directory_, Tan et al's Eclipse attack ([10.1109/JIOT.2018.2846624](https://doi.org/10.1109/JIOT.2018.2846624); reference 16 in this paper; no open access copy available). I am not qualified to evaluate the mathematical analysis in section 3, but it does seem to agree nicely with the experimental results in section 4. That experiment was well-designed and convincing, but with one major flaw: collateral damage should have been assessed (see questions section).

Writing quality is poor: excessively "purple" prose, making it difficult to understand what they actually did in several places. The framing of the paper is awkward, talking about "misuse of hidden services" for no apparent reason. This should be easy enough to correct.

There is no discussion of previous work on content suppression attacks on DHTs in general, which means that their suggested countermeasures are hard to assess.

**Questions:**

* You should have evaluated your attack's potential for collateral damage. That is, you should have presented both theoretical and experimental numbers for the probability that your attack will render _other_ hidden services inaccessible, besides the target(s). This is an ethical problem as well as a gap in your study, because you ran the attack on the real Tor network. It's possible that your 16 HSDirs held the only copies of the directory entries for legitimate services other than your test target. You had all the data you needed to detect and report this situation, but you didn't. This needs to be corrected before publication. Make sure to analyze the probability of collateral damage as a function of _how many services_ are being targeted for suppression -- in the field, there is never just _one_ DoS attack going on.

* You claim that, compared to Eclipse, "our method demonstrates a superior capability to accurately track the responsible HSDirs, rendering it [a more practical attack]".  Please provide concrete, quantitative evidence for this claim, e.g. by including Eclipse as a point of comparison in both the mathematical analysis (section 3) and the experiments (section 4). If this is not feasible in the time available for revisions, the claim should be dropped and replaced with a neutral comparison of your work to Eclipse.

* Please add to section 6 a review of content suppression attacks on DHTs in general; the basic form of your attack is well-known in the context of general-purpose distributed storage systems. This will give readers the context they need to understand why you suggested the countermeasures you did.

* Throughout, you have written what we call "purple prose" in English literary criticism -- using long, fancy words and complex sentences when short, common words and simple sentences will do. This makes it hard to understand what you did. Academic writing is supposed to be precise but it is also supposed to be clear. You should revise the entire paper with the goal of using short, common words as much as possible. In case you don't already know how to do that, I recommend the book "The Art of Readable Writing" by Rudolf Flesch.

### Post-Discussion Notes

Authors have addressed my ethical concerns and I'm happy to recommend publication, provided that the paper is revised to include their explanation of why collateral damage is extremely unlikely when only one service is targeted.

**Ethics Review Description:**

Carried out an experiment on the real Tor network with no consideration for collateral damage (see questions section of review for details)

**Ethics Review Flag:**

Yes

**Reviewer Confidence:**

3: The reviewer is confident but not certain that the evaluation is correct

**Scope:**

3: The work is somewhat relevant to the Web and to the track, and is of narrow interest to a sub-community

---

### Decision · Program_Chairs · 2024-01-22

**Decision:**

Accept

**Comment:**

This paper reveals a new practical DoS attack capable of blocking Tor hidden service directories. This is a "stealthy" attack, in that the targeted hidden service remains unaware of the attack.

 The reviewers' comments were mixed. On the one hand, they seemed to appreciate the novelty and impact of the proposed attack. At the same time, they noted a lack of comparison with previous work, including content supression attacks on DHTs in general. Additionally, important questions were raised on the theoretical cost estimation framework descried in the paper.

 After the authors provided a reponse, most reviewers found the clarifications useful. Among other items, the authors also provided a detailed response to questions around the cost estimation framework. Overall, the reviewers were willing to accept the paper, assuming the comments regarding the theoretical cost estimation framework are correctly addressed.

 ---